# Selective RyR2 inhibition reduces arrhythmia susceptibility in human cardiac slices

Micah K. Madrid[1], Katy A. Trampel[1], Batool Salman[1] (ID), Sharon A. George[1,2], Abigail N. Smith[3] (ID), Jeffrey N. Johnston[3] (ID), Tatiana Efimova[1] (ID), Bjorn C. Knollmann[4] (ID) and Igor R. Efimov[1,5]

[1] *Department of Biomedical Engineering, Northwestern University, Chicago, IL, USA*
[2] *Department of Pharmacology and Chemical Biology, University of Pittsburgh, Pittsburgh, PA, USA*
[3] *Department of Chemistry, Vanderbilt University, Nashville, TN*
[4] *Vanderbilt Center for Arrhythmia Research and Therapeutics, Vanderbilt University School of Medicine, Nashville, TN*
[5] *Department of Medicine, Northwestern University, Chicago, IL, USA*

Handling Editors: Natalia Trayanova & Haibo Ni

The peer review history is available in the Supporting Information section of this article (https://doi.org/10.1113/JP290283#support-information-section).

**Abstract figure legend** *ent*-Vert selectively inhibits RyR2-mediated sarcoplasmic reticulum (SR) Ca$^{2+}$ leak and prevents triggered activity in human cardiac slices. *A*, in human ventricular slices exposed to $\beta$-adrenergic stimulation with Iso and RyR2 sensitization with caffeine, RyR2 channels become hyperactive, leading to Ca$^{2+}$ sarcoplasmic leak and triggered electrical activity. *B*, *ent*-Vert binds to RyR2 and stabilizes its closed state, thereby reducing sarcoplasmic Ca$^{2+}$ leak and suppressing triggered electrical activity. These findings demonstrate the translational potential of selective RyR2 inhibition as a targeted antiarrhythmic strategy for ventricular arrhythmias.

**Micah Keanu Madrid** earned his PhD in Biomedical Engineering from George Washington University in 2024. He joined Northwestern University as a visiting scholar in 2022, transitioning to a postdoctoral scholar after completing his doctoral studies. His PhD research centred on novel multiparametric systems, while concurrently investigating physiological mechanisms related to *ent*-Vert applications. He currently contributes to clinical research on interventional transcatheter systems at Abbott, focusing on right-sided structural heart disease.

[Correction added on 21 April 2026 after first online publication: The author byline has been updated with authors "Abigail N. Smith" and "Jeffrey N. Johnston," who were inadvertently omitted from the previous version.]

**Abstract** Ryanodine receptor 2 (RyR2) hyperactivity is frequently observed in structural heart disease (SHD), commonly caused by ischemic heart disease. This aberrant $Ca^{2+}$ release promotes irregular electrical activity and life-threatening arrhythmias. Our previous work demonstrated that the pan-ryanodine receptor (RyR) inhibitor dantrolene can reverse arrhythmogenic substrates, but its lack of RyR2 selectivity limits its therapeutic potential. The unnatural verticilide enantiomer (*ent*-verticilide) was identified as a selective RyR2 inhibitor, showing promising specificity without activating skeletal muscle RyR1. This study evaluated the antiarrhythmic potential of a selective RyR2 antagonist, *ent*-(+)-verticilide (*ent*-Vert), in human ventricular slices. Right and left ventricular slices were prepared from human hearts not designated for transplantation. Pseudo-electrocardiograms (ECG) tracked premature ventricular contraction (PVC) incidence, and optical mapping identified arrhythmogenic substrates. Baseline optical recordings were obtained before treatment with isoproterenol (Iso) (250 nM) and caffeine (200 μM) to induce $Ca^{2+}$ leak and arrhythmogenic activity, followed by sequential *ent*-Vert application (1 and 3 μM). The effects of *ent*-Vert alone were tested with baseline recordings followed by a single 3 μM dose. Iso and caffeine increased PVC incidence in both right (RV) and left ventricular (LV) slices, but *ent*-Vert (1 and 3 μM) significantly reduced it. Iso and caffeine also shortened action potential duration (APD) in both slice types, but *ent*-Vert, after Iso and caffeine or alone (under control conditions), did not significantly change APD. In conclusion, *ent*-Vert suppressed arrhythmic triggers and reduced arrhythmogenic substrates in human ventricular slices, highlighting its potential as a targeted antiarrhythmic therapy for SHD.

(Received 7 October 2025; accepted after revision 5 January 2026; first published online 25 January 2026)

**Corresponding author** I. R. Efimov: Department of Biomedical Engineering, Northwestern University, 303 E Superior Street, Chicago, IL 60611, USA. Email: igor.efimov@northwestern.edu

## Key points

- RyR2 hyperactivity is a major driver of arrhythmias in structural heart disease, producing abnormal $Ca^{2+}$ release and premature ventricular contractions.
- The RyR2-selective inhibitor *ent*-Vert was evaluated in human RV and LV slices to determine its antiarrhythmic potential.
- $\beta$-Adrenergic and RyR2 activation with Iso + caffeine induced $Ca^{2+}$ leak, ectopic beats and shortened APD. Sequential *ent*-Vert application (1 and 3 μM) significantly suppressed ectopy incidence in a dose-dependent manner.
- *ent*-Vert did not alter APD or conduction velocity under control or stimulated conditions, demonstrating that it suppresses arrhythmic triggers without disrupting normal cardiac electrophysiology.

## Introduction

SHD encompasses a broad group of cardiac pathologies that alter the structure and function of the heart (Steinberg et al., 2010). These include ischemic cardiomyopathy, dilated cardiomyopathy, hypertrophic cardiomyopathy, infiltrative diseases such as amyloidosis, and valvular pathologies such as aortic and mitral stenosis (Watkins et al., 2017). Whether congenital or acquired, these abnormalities contribute substantially to global morbidity and mortality and can ultimately lead to sudden cardiac death (SCD) (El-Sherif et al., 2017; Gräni et al., 2020). Ventricular arrhythmias, including ventricular tachycardia and ventricular fibrillation, are triggers of SCD

(John et al., 2012). According to Coumel's triangle of arrhythmogenesis, three components are required for arrhythmia development: an arrhythmogenic substrate, a trigger and modulation by the autonomic nervous system, which can influence both the substrate and the triggering events (Cluitmans et al., 2023; Coumel, 1987).

Among SHDs, ischemic heart disease is a major contributor to arrhythmogenesis (Ghuran & Camm, 2001). It influences all three aspects of arrhythmogenesis: the arrhythmogenic substrate, triggers and autonomic modulation. Specifically, ischemic injury alters the substrate by changing APD and conduction velocity (CV), increases the likelihood of triggers through early and

delayed afterdepolarizations (EADs and DADs), and disrupts the autonomic nervous system and adrenergic signaling pathways (Cluitmans et al., 2023; Dries et al., 2020; Janse, 2004; Lazzara & Scherlag, 1984; Lopez & Malhotra, 2019). EADs result from changes during the repolarization phase of the cardiac action potential, whereas DADs stem from faulty $Ca^{2+}$ regulation during the heart's resting phase, causing abnormal $Ca^{2+}$ leakage (Amoni et al., 2021). Both types of afterdepolarizations can trigger PVCs, which may lead to arrhythmias (Tse, 2016).

At the cellular level, normal $Ca^{2+}$ handling starts with depolarization-triggered electrical excitation, which opens the L-type $Ca^{2+}$ channels ($Ca_v1.2$), allowing $Ca^{2+}$ to enter the cell (Bers, 2002). L-type $Ca^{2+}$ channels are mainly located at the junctions between the sarcolemma and the SR, where the SR $Ca^{2+}$ release channels, called RyRs, are present (Bers, 2002). The inward $Ca^{2+}$ influx through $Ca_v1.2$ enables $Ca^{2+}$ to bind to RyR2 on the SR, the $Ca^{2+}$ storage, leading to a significant release of $Ca^{2+}$ from the SR into the cell's interior – the process known as $Ca^{2+}$-induced $Ca^{2+}$ release (CICR) (Bers, 1993, 2002). The released cytosolic $Ca^{2+}$ then binds to troponin C, initiating cardiac muscle contraction (Gordon et al., 2000). Afterwards, $Ca^{2+}$ is removed from the cell's interior either back into the SR via the sarcoplasmic/endoplasmic reticulum $Ca^{2+}$-ATPase 2A (SERCA2a) and out of the cell through the $Na^+/Ca^{2+}$ exchanger (NCX) (Blaustein & Lederer, 1999). However, in various SHDs, RyR2 can become hyperactive (Dries et al., 2018; Fauconnier et al., 2011; Li et al., 2014; Terentyev et al., 2008). This results in unintended $Ca^{2+}$ leakage from the SR during the heart's relaxation phase (diastole), independent of normal excitation–contraction coupling. This phenomenon has been observed in several human heart diseases associated with atrial and ventricular arrhythmias (Yano et al., 2005; Bers, 2006).

Previously, we found that the arrhythmogenic state of SHD in the human heart, simulated by $\beta$-adrenergic and RyR2 activation, could be suppressed by acute inhibition of RyR with dantrolene (George et al., 2023). However, dantrolene is a pan-inhibitor of RyR, meaning it targets all isoforms. This limits its clinical use for treating cardiac arrhythmias because it causes off-target effects such as reduced peripheral muscle strength and hepatotoxicity with long-term use (Krause et al., 2004). In humans, there are three isoforms of RyR: RyR1, RyR2 and RyR3. RyR2 is the predominant cardiac isoform (Kistamás et al., 2020). The unnatural enantiomer of the fungal product verticilide – *ent*-Vert – selectively inhibits RyR2 with nanomolar potency, has no off-target effects on cardiac membrane ion channels, and has no activity on skeletal muscle RyR1 (Batiste et al., 2019; Blackwell et al., 2023). Here, we utilized *ent*-Vert to test the effect of selective RyR2 inhibition on cardiac electrophysiology

and arrhythmogenesis in human heart models of SHD.

A combination of pharmacological $\beta$-adrenergic stimulation (Iso) and activation of RyR2 (caffeine) was employed to simulate the abnormal autonomic nervous system activity and $Ca^{2+}$ dysregulation observed in hearts affected by SHD in human organotypic LV and RV slices. Administration of *ent*-Vert reversed the formation of arrhythmogenic substrates and triggers, demonstrating its antiarrhythmic potential.

## Methods

### Ethical approval

Human donor hearts not used for transplantation were acquired through our collaboration with the organ procurement organizations Gift of Hope (Chicago, IL, USA) and Novabiosis (Indianapolis, IN, USA). The study protocol, submitted to Northwestern University's Institutional Review Board (IRB), was deemed IRB exempt (IRB ID: STU00216867). Patient demographics of the nine hearts used in this study are listed in Table 1. These comorbidities, including hypertension and high cholesterol, are highly prevalent in the population that requires antiarrhythmic therapy, and therefore the donor hearts used in this study accurately represent this population.

### Ventricular slice preparations

The LV and RV samples were isolated from the anterior sides of the free wall of the base using transmural incisions measuring 1 cm by 1 cm apart. The excised tissue was placed on a tissue holder of the precision vibrating micro-tome (Campden Instruments, Loughborough, United Kingdom, 7000-smz series). The epicardial surface was adhered to the holder using Histoacryl skin adhesive, with a layer of 4% agarose perpendicular to the cutting direction. The slicing process was carried out at a frequency of 80 Hz, with a thickness of 400 μm, an oscillation amplitude of 2 mm and an advance rate of 0.04 mm/s. This was done in a modified Tyrode's solution that included 140 mM NaCl, 6 mM KCl, 1 mM $MgCl_2$, 1.8 mM $CaCl_2$, 10 mM glucose, 10 mM HEPES, and 10 mM 2,3-butanedionemonoxime (BDM) at a pH of 7.4, all on ice. After slicing, the slices were incubated in a recovery solution with a similar composition but with 4.5 mM KCl instead of 6 mM, at room temperature for at least 20 min before proceeding to optical mapping.

### Optical mapping

Slices from the LV and RV, as shown in Fig. 1A, were secured onto a layer of polydimethylsiloxane inside a

**Table 1. Summary of human donor heart demographics**

| Age (years) | Sex | Cause of death | EF (%) | Comorbidities | Preparation | BMI |
|---|---|---|---|---|---|---|
| 51 | F | CVA/stroke | N/A | N/A | LV/RV | 31.2 |
| 42 | M | CVA/stroke | N/A | Hypertension, hyperlipidemia, congestive heart failure, coronary artery disease, chronic pancreatitis, kidney stones, chronic kidney disease, acute kidney injury | LV | 22 |
| 25 | M | CVA/stroke | N/A | N/A | RV | 28.8 |
| 54 | F | CVA/stroke | N/A | N/A | LV/RV | 35.1 |
| 38 | M | Head trauma | 64 | Hypertension, hypercholesterolaemia | LV/RV | 26.5 |
| 64 | F | CVA/stroke | 62 | N/A | LV/RV | 28.8 |
| 49 | F | CVA/stroke | 60 | Hypertension | LV/RV | 31.7 |
| 45 | M | CVA/stroke | NA | Hypertension, enlarged heart, pacemaker | RV | 27.5 |
| 62 | M | Anoxia | 65 | Hypertension, diabetes | LV | 26.2 |

BMI, body mass index; CVA, cerebrovascular accident; EF, ejection fraction.

tissue bath (AD Instruments, Colorado Springs, CO, USA, Radnotti 1584 series) and continuously superfused with a recovery solution enhanced with 15 μM blebbistatin (Cayman Chemical Co., Ann Arbor, MI, USA, 13186), maintained at 37°C. Electrical stimulation was applied to pace the tissue with a central bipolar pacing electrode. The stimulation was set at 1.5 times the threshold required for pacing capture, with a pulse duration of approximately 2 ms, and basic cycle lengths (BCLs) varied.

The experimental protocol, shown in Fig. 1*B*, started with an initial equilibration phase before baseline measurements. It was then followed by adding 250 nM Iso (MilliporeSigma, Burlington, MA, USA, I5627) and 200 μM caffeine (MilliporeSigma, C0750) to stimulate $\beta$-adrenergic receptors and induce hyperactivity in RyR2. Next, *ent*-Vert was applied at different concentrations (1 and 3 μM) to inhibit RyR2 and to assess dose-dependent effects. Although these interventions do not fully replicate all aspects of SHD, they help simulate conditions of increased sympathetic activity and $Ca^{2+}$ mishandling due to leaky RyR2, as previously described (George et al., 2023). The study evaluates *ent*-Vert's potential as an antiarrhythmic agent under these conditions.

RH237 (Biotium, Fremont, CA, USA, 61018) was used with excitation at 520 nm to record transmembrane potential. Data were collected using $100 \times 100$ MiCAM Ultima CMOS cameras (SciMedia, Costa Mesa, CA, USA) with an approximately $27 \times 27$ mm$^2$ field of view to accommodate two slices simultaneously. The data were then processed using Rhythm 3.0 software to analyse APD and transverse conduction velocity ($CV_T$) over specific areas of the tissue, with values averaging over at least three cycles.

### Pseudo-ECG

ECG electrodes were positioned in the tissue bath to capture pseudo-ECGs from the LV and RV slice setups. These ECG signals were recorded at a -kHz sampling rate using PowerLab 4/26 and LabChart software (ADInstruments, Colorado Springs, CO, USA) The ECG data were then analysed to identify ectopic beats. Ectopic beats were defined as those not triggered by pacing and originating from a different location than the pacing site, as confirmed through optical mapping, and showing a unique ECG morphology compared to paced beats.

### Statistics

Data are presented as means ± standard deviation unless specified otherwise. Statistical analyses were conducted using GraphPad Prism software (GraphPad Software, Boston, MA, USA). The Shapiro–Wilk test assessed the normality of data sets. For data that were not normally distributed, such as ectopy incidence in Fig. 2*C*, Friedman's test was used to identify significant differences among groups, followed by Wilcoxon's signed-rank *post hoc* test for detailed comparisons. For normally distributed data like LV and RV APD at 1-s basic cycle length in Fig. 3*C*, a one-way ANOVA or a mixed effects model (depending on missing values) with repeated measures was used to detect differences, with post hoc analysis via Tukey's multiple comparison test. Restitution data were analysed using non-linear regression with an exponential plateau model, ensuring a good fit with $r^2 > 0.5$ and verifying the randomness in residual plots. Significance was set at $P < 0.05$.

## Results

### RyR2 inhibition reduces the incidence of ventricular ectopy in human hearts

Optical mapping of slice preparations revealed distinct activation patterns between paced and ectopic beats, as shown in Fig. 2*A* and *B*. In Fig. 2*A*, a paced beat (red line) originates at the pacing site (white star), while an ectopic beat (green line) arises from a different location, distant from the pacing site. Figure 2*B* illustrates these differences across four conditions: (1) baseline, with a paced beat originating at the pacing site (shown on the figure by a white star); (2) treatment with Iso + caffeine, also showing a paced beat from the pacing site; (3) an ectopic beat under Iso + caffeine treatment, originating away from the pacing site with a markedly different activation time range; and (4) treatment with Iso + caffeine + *ent*-Vert (1 μM), reverting to a paced beat originating at the pacing site. These images highlight the shift in activation origin and timing for ectopic beats compared to paced beats under different treatment conditions.

Slice preparations from the LV and RV of the same heart (*n* = 4 hearts) were continuously paced at 1 Hz throughout the experiment, with pseudo-ECGs recorded and monitored. During baseline treatment, no ectopic beats were observed in the slices from either ventricle. In subsequent tests, treatments with Iso + caffeine, Iso + caffeine + *ent*-Vert 1μM, and Iso + caffeine + *ent*-Vert 3μM resulted in average ectopy incidence rates of 39.38, 20.16 and 9.38 PVC per minute (PVC/min), respectively, when combining data from both ventricles (Fig. 2*C*). Notably, the LV exhibited higher ectopy rates of 50.63, 25.31 and 18.75 PVC/min, while the RV showed rates of 28.13, 15.00 and 0.00 PVC/min, respectively. In some hearts, ectopy was observed in the LV but absent in the RV under the same treatment conditions. The overall average reflects these combined LV and RV responses. *ent*-Vert showed a trend toward a reduction in the incidence of Iso + caffeine-induced ectopy, indicating suppression of $Ca^{2+}$-mediated triggers.

### Effects of *ent*-Vert on electrical arrhythmogenic substrate in LV and RV slices

This study utilized optical mapping of transmembrane potential to examine the effects of *ent*-Vert on electrical arrhythmogenic substrates in LV and RV slices. The slices were tested under baseline conditions, Iso + caffeine, and

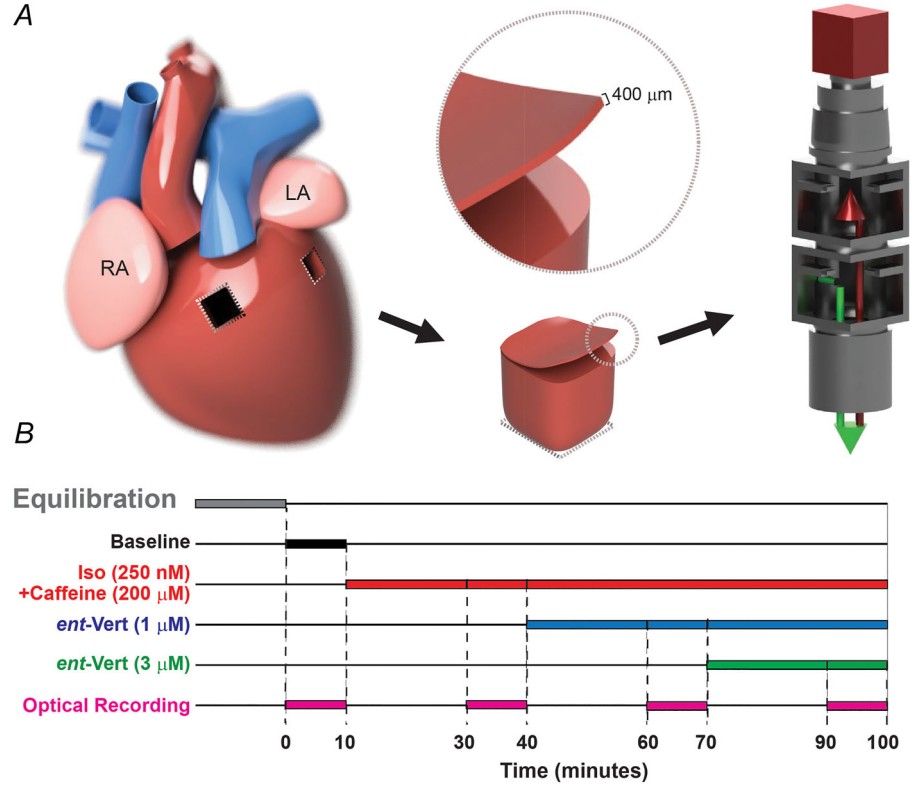

**Figure 1. Schematic illustration of human donor heart organotypic slice preparation and optical mapping protocol**
*A*, organotypic slices were prepared from the base of the RV and LV free walls, sectioned to 400 μm thickness, and used for optical mapping of transmembrane potential. *B*, experimental timeline and mapping protocol.

Iso + caffeine + *ent*-Vert treatments. Representative OAP traces from LV and RV under these conditions are shown in Fig. 3*A*, with corresponding APD maps in Fig. 3*B*.

Iso + caffeine treatment significantly reduced APD in both LV and RV, as demonstrated by restitution curves and 1-s measurements in Fig. 3*C* and *D*. At 1-s pacing, baseline APD values were 499.9 ± 38.9 ms (LV) and 440.7 ± 81.1 ms (RV). After treatment with Iso + caffeine, APD decreased to 331.4 ± 102.3 ms (LV) and 318.8 ± 77.87 ms (RV) ($P < 0.05$). Adding 1 μM *ent*-Vert resulted in an APD of 302.7 ± 47.5 ms (LV) and 283.8 ± 63.3 ms (RV), while 3 μM *ent*-Vert yielded 305.0 ± 27.2 ms (LV) and 322.1 ± 44.67 ms (RV). These changes were not statistically significant across multiple comparisons ($P > 0.05$), but they were significant compared to baseline, indicating that the main effects are due to Iso + caffeine. The overall difference was significant ($P < 0.05$, ANOVA). Additionally, the ANOVA/mixed effects test was not significant for $CV_T$, although multiple comparisons showed some significant differences between baseline and Iso + caffeine as seen in Fig. 3*E* and *F*.

In summary, Iso + caffeine treatment induced arrhythmogenic triggers (as previously reported) and established an arrhythmogenic substrate by significantly

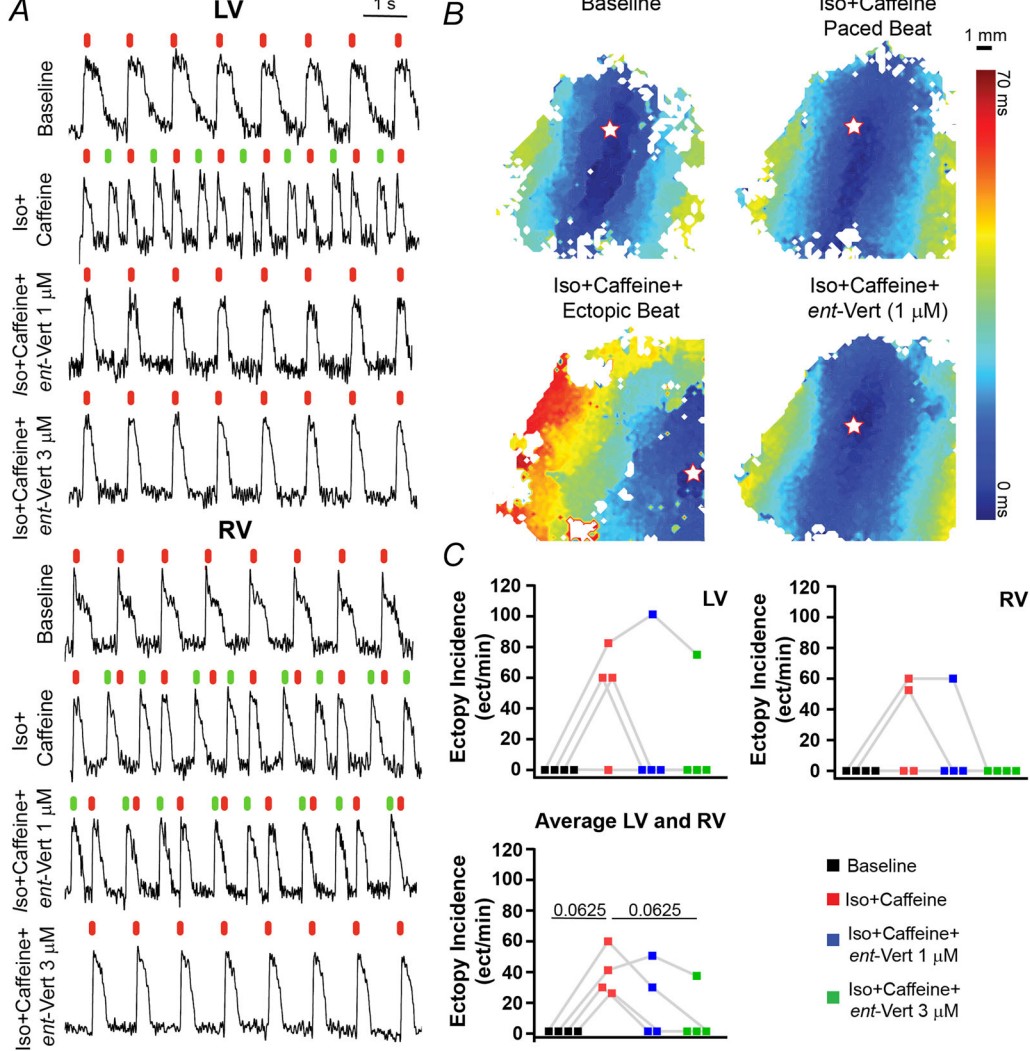

**Figure 2. RyR2 inhibition with *ent*-Vert suppresses ectopic activity in human hearts**
*A*, optical action potential (OAP) traces show paced beats (aligned with red pacing stimuli) and unpaced ectopic beats (indicated by a green line) from the LV (top) and RV (bottom) slices during baseline, Iso + caffeine, and Iso + caffeine + *ent*-Vert treatments. *B*, optical activation maps of paced *versus* unpaced beats illustrate different origin sites (marked with a star). Top left: baseline; top right: Iso + Caffeine paced beat; bottom left: Iso + Caffeine ectopic beat; bottom right: *ent*-Vert paced beat. *C*, summary of ectopic incidence rates in LV and RV slice preparations and averaged per heart. Sample sizes: RV: *n* = 4; LV slices: *n* = 4. Friedman's test was used to identify significant differences between experimental groups, with Wilcoxon's signed-rank test as a *post hoc* analysis to determine differences in ectopy incidence between specific treatments.

shortening APD in LV and RV slices, mimicking conditions seen in SHD. Notably, *ent*-Vert did not significantly change APD but suppressed arrhythmogenic triggers, suggesting its potential as an antiarrhythmic therapy through mechanisms independent of APD modulation.

### Isolated effects of *ent*-Vert in LV and RV slices

Optical mapping of transmembrane potential was performed to assess the isolated effects of *ent*-Vert on LV and RV slices. Representative OAPs from LV and RV slices under baseline conditions and after treatment with 3 μM *ent*-Vert are shown in Fig. 4*A*, with corresponding APD maps displayed in Fig. 4*B*. APD and $CV_T$ in LV and RV slices remained unchanged from baseline and after

*ent*-Vert administration (Fig. 4*C* and *D*). These findings suggest that *ent*-Vert does not significantly affect APD or $CV_T$ in LV and RV slices, supporting its potential safety and suitability as an antiarrhythmic therapy.

## Discussion

This study evaluated *ent*-Vert, a selective RyR2 inhibitor, for its effects on arrhythmia susceptibility in LV and RV slices from human hearts under conditions that mimic SHD. Our findings showed that *ent*-Vert significantly suppressed arrhythmogenic triggers (Fig. 2*C*), evidenced by a dose-dependent decline in ectopic beat incidence, while not significantly affecting APD or $CV_T$ (Figs 3*D* and *F*, 4*E* and *F*). These results indicate that *ent*-Vert is a promising candidate for targeted antiarrhythmic therapy

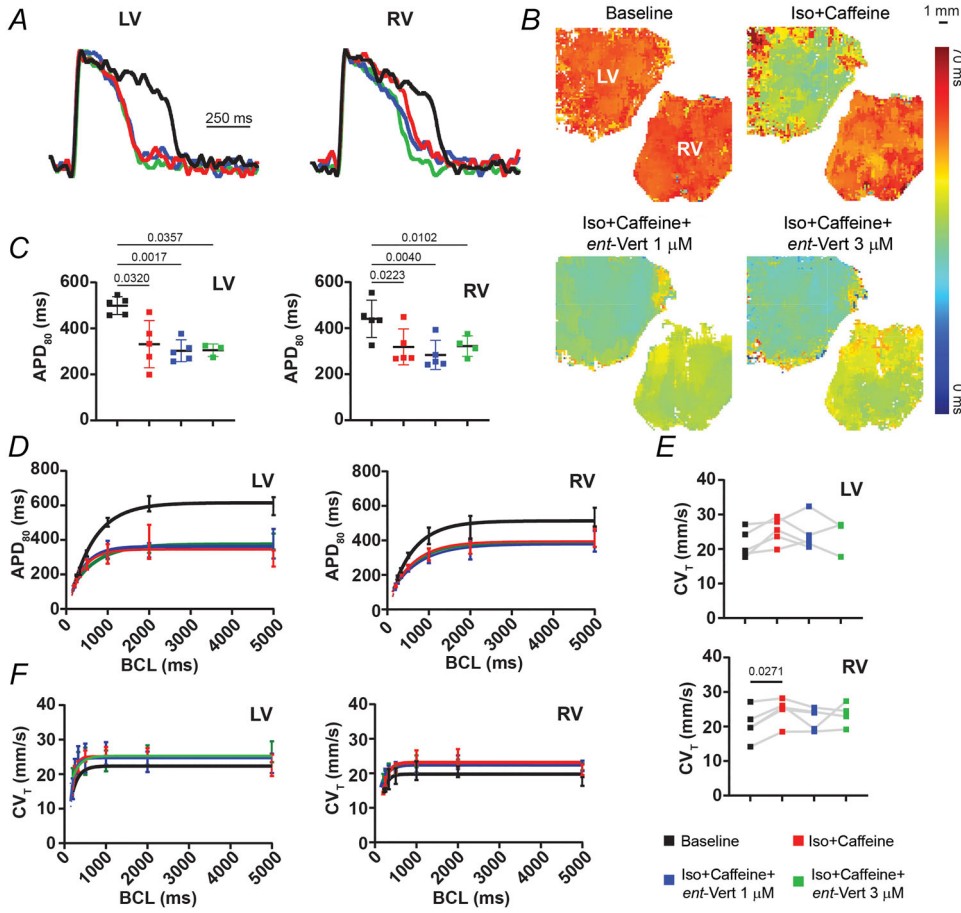

**Figure 3. RyR2 inhibition does not alter APD in LV or RV slices**
*A*, representative OAP traces during 1-s BCL pacing. *B*, APD maps from LV and RV slices. *C*, average APD during 1-s BCL pacing under baseline, Iso + Caffeine, and Iso + Caffeine + *ent*-Vert treatments from LV and RV slices. *D*, APD restitution curves for each treatment condition. *E*, average $CV_T$ during 1-s BCL pacing across treatment conditions in LV and RV slices. *F*, $CV_T$ restitution curves during each treatment condition. A sample size of 5 LV and RV slices was used. For data in *C* and *E*, a one-way repeated-measures ANOVA followed by Tukey's multiple comparisons test for *post hoc* analysis was performed for data at 1-s BCL. For data in *D* and *F*, nonlinear regression analysis using the least squares method and exponential plateau model was performed to determine significant differences in restitution curves between treatments. Iso, isoproterenol; *ent*-Vert, *ent*-verticilide.

in SHD, potentially offering advantages over less selective RyR inhibitors like dantrolene.

$Ca^{2+}$ is essential for cardiac contraction and cardiomyocyte function. It plays a crucial role in mitochondrial adenosine triphosphate (ATP) production, which influences the energy supply for heart activities (Bers & Grandi, 2009; Peracchia, 2020; Rhana et al., 2024). $Ca^{2+}$ also regulates the gating of ion channels and gap junctions, potentially leading to abnormal APD and electrical propagation, which can contribute to arrhythmogenesis (Bers & Grandi, 2009; Peracchia, 2020). This ion's regulation also involves metabolism–excitation–contraction (MEC) coupling, ensuring coordinated cardiac function (Eisner et al., 2017). However, disruptions in $Ca^{2+}$ homeostasis can lead to MEC uncoupling, fostering conditions that promote arrhythmias (Nattel et al., 2007; Li et al., 2014).

Cardiac arrhythmias in SHD are a leading cause of morbidity and mortality, driven by pathological remodelling that disrupts $Ca^{2+}$ homeostasis (Bers, 2008). Hyperactive RyR2 channels cause diastolic $Ca^{2+}$ leaks, leading to DADs and ectopic beats (Wehrens et al., 2006). *ent*-Vert's selective inhibition of RyR2 resulted in a reduction in ectopy rates, with complete suppression in the RV at 3 μM (Fig. 2C, top right), while the LV showed partial suppression (Fig. 2C, top left). This chamber-specific response suggests possible differences in RyR2 expression or function, warranting further investigation.

The antiarrhythmic effect of *ent*-Vert likely results from its ability to inhibit RyR2-mediated $Ca^{2+}$ leaks, a key factor in triggering arrhythmias in SHD. Consistent with its selective RyR2-modulating mechanism, *ent*-Vert did not significantly alter APD in human ventricular slices, while dantrolene, a pan-RyR inhibitor, has previously been observed to modestly prolong APD in human ventricular slices (George et al., 2023). This suggests that, at these doses, its antiarrhythmic action targets triggers rather than the electrophysiological substrate. Similarly, $CV_T$ remained unchanged, with no significant differences seen in either combined or separate treatment conditions. The overall ANOVA for APD was significant ($P < 0.05$), but the ANOVA/mixed effects test for CV was not. However, multiple comparisons revealed some significant differences between baseline and Iso + caffeine, further indicating that *ent*-Vert's effects at these concentrations are more specific to arrhythmia triggers.

Unlike studies using dual voltage and $Ca^{2+}$ optical mapping, such as George et al. (2023), which employed Rhod2-AM for $Ca^{2+}$ imaging in LV slices, our protocol relied solely on transmembrane potential mapping. This choice was based on concerns that $Ca^{2+}$ dyes like Rhod2-AM might buffer intracellular $Ca^{2+}$, potentially altering $Ca^{2+}$ dynamics and decreasing ectopic beat incidence. Although research in human-induced pluripotent stem cell-derived cardiomyocytes suggests that $Ca^{2+}$-sensitive dyes have minimal direct effects on electrophysiological properties (Kopljar et al., 2018), their

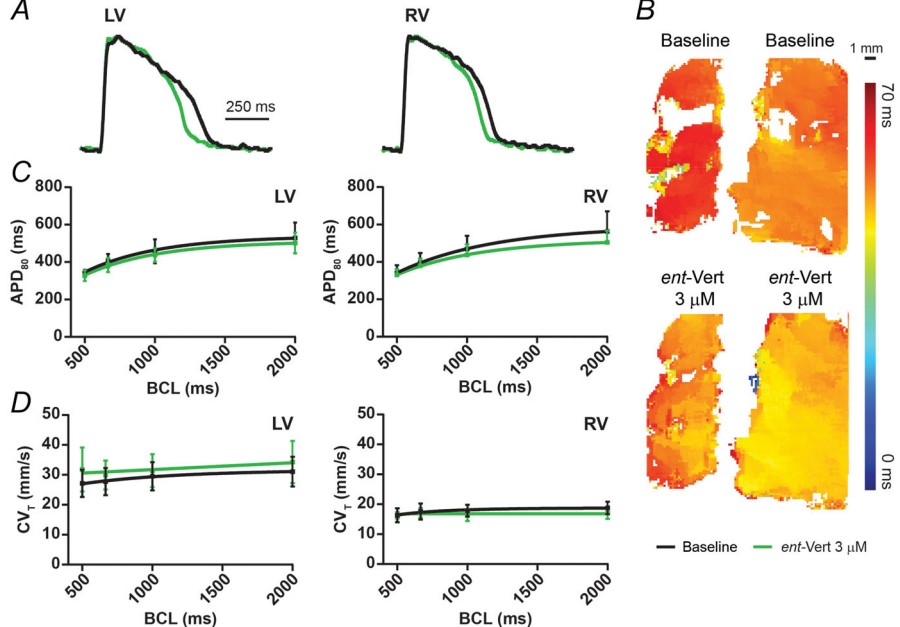

**Figure 4. Isolated effects of RyR2 inhibition with *ent*-Vert in LV and RV slices**
*A*, representative OAP traces during 1-s BCL pacing. *B*, APD maps from LV and RV slices. *C*, APD restitution curves under baseline and *ent*-Vert (3 μM) treatment in LV and RV slices. *D*, $CV_T$ restitution curves for each treatment condition in LV and RV slices. For data in *C* and *D*, nonlinear regression analysis with the least squares method and exponential plateau model was used to identify significant differences in restitution curves between treatments.

effects on adult human ventricular slices are less well understood. By omitting $Ca^{2+}$ dyes, we ensured that the observed reduction in ectopy was solely due to *ent*-Vert, avoiding possible confounding effects. Future research, including $Ca^{2+}$ imaging, could clarify how *ent*-Vert influences $Ca^{2+}$ handling to inhibit arrhythmias.

In isolated treatments, *ent*-Vert (3 μM) caused non-significant changes in APD and $CV_T$ in both LV and RV slices (Fig. 4*E* and *F*), underscoring its specificity and potential safety. Unlike dantrolene, which affects RyR1 in skeletal muscle and can cause side effects like muscle weakness (Krause et al., 2004), *ent*-Vert's selectivity for RyR2, predominantly expressed in cardiac tissue, may reduce such risks, enhancing its suitability for clinical development.

Our findings align with previous research demonstrating *ent*-Vert's antiarrhythmic effects in animal models of catecholaminergic polymorphic ventricular tachycardia (CPVT). Studies in Casq2$^{-/-}$ mice showed that *ent*-Vert reduces RyR2-mediated $Ca^{2+}$ leaks and prevents ventricular arrhythmias [29]. This study extends these findings to human cardiac tissue, providing crucial evidence for its translational potential in SHD. The absence of significant changes in APD or $CV_T$ contrasts with dantrolene's broader electrophysiological effects, indicating that *ent*-Vert might be a more targeted treatment.

### Limitations and future directions

We acknowledge several limitations. The study used *ex vivo* human ventricular slices, which may not fully mimic *in vivo* cardiac function. The smaller sample sizes could limit statistical power, but the changes in ectopy incidence observed with Iso and caffeine treatments indicate that the variability-to-effect ratio was sufficient to detect key electrophysiological responses. This supports the robustness of these findings despite the limited cohort. Additionally, the chamber-specific differences in ectopy suppression (complete in the RV *versus* partial in the LV at 3 μM) suggest possible regional variations in RyR2 expression or function, which require further investigation. Future studies should explore *ent*-Vert's long-term effects, its efficacy in whole-heart or *in vivo* models of SHD, and its pharmacokinetic profile in humans, building on evidence from murine studies showing a half-life of 6–7 h (Gochman et al., 2023).

### Conclusion

*ent*-Vert shows significant potential as an antiarrhythmic agent by selectively targeting RyR2-mediated $Ca^{2+}$ leaks in human heart tissue, effectively reducing ectopic beats without altering APD or CV. These findings highlight its promise as a targeted therapy for arrhythmias in SHD and may offer advantages over less selective inhibitors like dantrolene. Further research is needed to confirm these effects in larger studies and *in vivo* models, paving the way for clinical development of *ent*-Vert as a new anti-arrhythmic treatment.

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

## Additional information

### Data availability statement

Data supporting the findings of this study are available from the corresponding author upon request.

### Competing interests

The authors declare no conflicts of interest, financial or otherwise.

### Author contributions

Conceived and designed experiments – M.K.M., S.A.G., I.R.E.; performed experiments – M.K.M., K.A.T., B.S., S.A.G.; analysed data – M.K.M., K.A.T.; interpreted results of experiments – M.K.M., K.A.T., I.R.E.; prepared figures – M.K.M., K.A.T.; provided resources (synthesis of the experimental study drug ent-verticilide) – A.N.S., J.N.J.; funding acquisition – J.N.J.; drafted manuscript – M.K.M., K.A.T., I.R.E.; edited and revised manuscript – K.A.T., M.K.M., T.E., B.C.K., I.R.E. All authors have read and approved the final version of this manuscript and agree to be accountable for all aspects of the work in ensuring that questions related to the accuracy or integrity of any part of the work are appropriately investigated and resolved. All persons designated as authors qualify for authorship, and all those who qualify for authorship are listed.

[Correction added on 6 May 2026, after first online publication: "The Author Contribution section has been updated to include the contribution of two authors Abigail N. Smith and Jeffrey N. Johnston."]

### Funding

This work was supported by the American Heart Association (AHA) Sudden Cardiac Death SFRN Grant 19SFRN34830033 (to B.K. and I.R.E.).

### Acknowledgements

We are grateful to the donor families and to the organ procurement teams at Novabiosis and Gift of Hope for their contributions and assistance in providing human heart tissues.

### Keywords

ryanodine receptor 2, RyR2, *ent*-verticilide, structural heart disease

## Supporting information

Additional supporting information can be found online in the Supporting Information section at the end of the HTML view of the article. Supporting information files available:

**Peer Review History**

