## [Peer Review History · The Journal of Physiology]

Selective RyR2 Inhibition Reduces Arrhythmia Susceptibility in Human Cardiac Slices

Micah Madrid, Katy Trampel, Batool Salman, Sharon A George, Tatiana Efimova, Bjorn C. Knollmann, and Igor R. Efimov
DOI: 10.1113/JP290283

Corresponding author(s): Igor Efimov (igor.efimov@northwestern.edu)

The following individual(s) involved in review of this submission have agreed to reveal their identity: David A Eisner (Referee #1); Dobromir Dobrev (Referee #2)

Review Timeline:

Submission Date:	07-Oct-2025
Editorial Decision:	29-Oct-2025
Revision Received:	02-Dec-2025
Accepted:	05-Jan-2026

Senior Editor: Natalia Trayanova

Reviewing Editor: Haibo Ni

Transaction Report:

Re: JP-RP-2025-290283 "**Selective RyR2 Inhibition Reduces Arrhythmia Susceptibility in Human Cardiac Slices**" by Micah Madrid, Katy Trampel, Batool Salman, Sharon George, Tatiana Efimova, Bjorn C. Knollmann, and Igor R. Efimov

Dear Dr Efimov,

Thank you for submitting your manuscript to The Journal of Physiology. It has been assessed by a Reviewing Editor and by 2 expert referees and we are pleased to tell you that it is potentially acceptable for publication following satisfactory major revision.

Please address all the points raised and incorporate all requested revisions or explain in your Response to Referees why a change has not been made. We hope you will find the comments helpful and that you will be able to return your revised manuscript within 2 months. If your article is NOT for a Special Issue, you may have 9 months to revise. If you require an extension, please contact journal staff: jp@physoc.org. Please note that this letter does not constitute a guarantee for acceptance of your revised manuscript.

REVISION CHECKLIST:

We look forward to receiving your revised submission.

Yours sincerely,

Natalia Trayanova
Senior Editor
The Journal of Physiology

REQUIRED ITEMS FOR REVISION

- Author photo and profile. First or joint first authors are asked to provide a short biography (no more than 100 words for one author or 150 words in total for joint first authors) and a portrait photograph. These should be uploaded and clearly labelled together in a Word document with the revised version of the manuscript. See Information for Authors for further details.

- You must start the Methods section with a paragraph headed Ethical Approval. If experiments were conducted on humans, confirmation that informed consent was obtained, preferably in writing, that the studies conformed to the standards set by the latest revision of the Declaration of Helsinki and that the procedures were approved by a properly constituted ethics committee, which should be named, must be included in the article file. If the research study was registered (clause 35 of the Declaration of Helsinki), the registration database should be indicated, otherwise the lack of registration should be noted as an exception (e.g. The study conformed to the standards set by the Declaration of Helsinki, except for registration in a database). For further information see: <https://physoc.onlinelibrary.wiley.com/hub/human-experiments>.

- The reference list must be in alphabetical order, rather than numbered, to comply with our Journal format.

- Please upload separate high-quality figure files via the submission form.

- Please ensure that the Article File you upload is a Word file.

- Papers must comply with the Statistics Policy: https://jp.msubmit.net/cgi-bin/main.plex?form_type=display_requirements#statistics.

In summary:

- If $n \leq 30$, all data points must be plotted in the figure in a way that reveals their range and distribution. A bar graph with data points overlaid, a box and whisker plot or a violin plot (preferably with data points included) are acceptable formats.

- If $n > 30$, then the entire raw dataset must be made available either as supporting information, or hosted on a not-for-profit repository, e.g. FigShare, with access details provided in the manuscript.

- 'n' clearly defined (e.g. x cells from y slices in z animals) in the Methods. Authors should be mindful of pseudoreplication.

- All relevant 'n' values must be clearly stated in the main text, figures and tables.

- The most appropriate summary statistic (e.g. mean or median and standard deviation) must be used. Standard Error of the Mean (SEM) alone is not permitted.

- Exact p values must be stated. Authors must not use 'greater than' or 'less than'. Exact p values must be stated to three significant figures even when 'no statistical significance' is claimed.

- Please include an Abstract Figure file, as well as the Figure Legend text within the main article file. The Abstract Figure is a piece of artwork designed to give readers an immediate understanding of the research and should summarise the main conclusions. If possible, the image should be easily 'readable' from left to right or top to bottom. It should show the physiological relevance of the manuscript so readers can assess the importance and content of its findings. Abstract Figures should not merely recapitulate other figures in the manuscript. Please try to keep the diagram as simple as possible and without superfluous information that may distract from the main conclusion(s). Abstract Figures must be provided by authors no later than the revised manuscript stage and should be uploaded as a separate file during online submission labelled as File Type 'Abstract Figure'. Please also ensure that you include the figure legend in the main article file. All Abstract Figures should be created using BioRender. Authors should use The Journal's premium BioRender account to export high-resolution images. Details on how to use and access the premium account are included as part of this email.

- Please include a full title page as part of your main article (Word) file, which should contain the following: title, authors, affiliations, corresponding author name and contact details, keywords, and running title.

- Please ensure that all figures and tables have a title and legend, and that they have been cited within the main article text.

EDITOR COMMENTS

Reviewing Editor: Ethics Concerns:

This study involves human donor hearts that were not used for transplantation, and the study protocol, submitted to local (Northwestern University's) Institutional Review Board (IRB), was deemed IRB exempt. It might not be applicable/possible to have written consents in this case.

An appropriate reference number for the approval is not stated.

Comments to the Author:

The manuscript has been reviewed by experts in this field. The major strengths of the manuscript include testing whether selective RyR2 inhibition is effective against arrhythmias, and use of human cardiac tissue. However, the reviewers also noted several weaknesses that require attention. These pertain to the arrhythmia induction protocol relying on caffeine, lack of experimental data at the cardiomyocyte level, some aspects of data interpretation, and statistical approaches and power. The reviewing editor concurs with these assessments.

REFEREE COMMENTS

Referee #1:

Madrid et al have investigated the effects of ent-verticilide on some factors underpinning arrhythmias. They conclude that the drug suppresses arrhythmic triggers and reduces arrhythmogenic substrates. Unlike dantrolene, verticilide does not affect skeletal muscle potentially making it an attractive choice as an antiarrhythmic.

A major strength of the work is that it is carried out on human cardiac tissue. There are, however, several weaknesses as described below.

1. Ectopy is induced by adding iso together with caffeine. The use of caffeine means that any arrhythmias arise, at least in part, from increasing the open probability of the RyR, This makes it less surprising that a drug which decreases RyR2

opening can decrease ectopy. It would have been more interesting if the arrhythmias had been provoked without interfering with the RyR.

2. While it is good that you test the data for normality before using parametric tests, it should be noted that such tests have little power to reveal deviations from normality with the relatively small sample sizes used.
3. You use Dunn's post hoc test in Fig 2C. As discussed in the original paper (Dunn, 1964) this assumes that the populations are identically distributed. This is not the case in your data where there is zero spread in the baseline condition. I suggest that you check with a statistician whether this may invalidate the apparent significance of your result.
4. In Fig 2C there do not appear to be 4 points in each of the conditions. I assume that some are superimposed but this needs to be made clearer, perhaps by separating horizontally.
5. Lines 220-222. You state that in some hearts ectopy was observed in the LV but not in the RV. However, the two upper panels of Fig 2C show 4 points with ectopy in the LV and 3 in the RV. This doesn't seem like a solid base from which to infer chamber differences. Is there any statistical basis to the statement?
6. Lines 277-280. Here you compare the lack of effect of ent-Vert which does not affect APD with dantrolene. You quote your previous work (George et al) as showing that dantrolene "tends to prolong APD in LV slices". It does not seem rigorous to compare non-significant effects of two drugs in this way.
7. Given the lack of effect on APD, why do you conclude (line 53) that ent-Vert "reduced arrhythmogenic substrates".

Referee #2:

The paper addresses one important aspect, namely whether selective RyR2 inhibition is effective against arrhythmias. The used novel and highly selective RyR2 inhibitor was shown to be effective in genetically modified mice with an increased susceptibility to arrhythmias. Since this cannot be directly translated to the human, such studies and the present one are essential for successful clinical translation.

The authors show in human ventricular slices that ventricular ectopy induced by ISO+caffeine is significantly reduced by the RyR2 inhibitors. CV and APD remained unchanged. The latter was interpreted as justification for the claim that only focal ectopic activity, but reentry termination is the potential reason for the antiarrhythmic effect of the RyR2 inhibitor.

I have the following comments:

The author studied the effects in tissue. How sure are you that the penetration of the drug was sufficient enough to allow to conclude that CV and APD are reliably UNchanged? Could APD recordings be repeated in isolated human ventricular myocytes?

The effect of the RyR2 inhibitor on RyR2 function is rather indirect. Any chance to show more direct effects (RyR2 single channel recordings or CaT properties, SR Ca²⁺ leak at the single cardiomyocyte level).

The effects shown with this new drug are highly variable in Fig. 2, with absence of effect in some preparations. Could this be a bias? Did the authors calculate the required sample size to reach reliable statistical conclusions?

END OF COMMENTS

REQUIRED ITEMS FOR REVISION

- Author photo and profile. First or joint first authors are asked to provide a short biography (no more than 100 words for one author or 150 words in total for joint first authors) and a portrait photograph. These should be uploaded and clearly labelled together in a Word document with the revised version of the manuscript. See https://jp.msubmit.net/cgi-bin/main.plex?form_type=display_requirements#authorprofile Information for Authors for further details.

Biography:

Micah Keanu Madrid earned his PhD in Biomedical Engineering from George Washington University in 2024. He joined Northwestern University as a visiting scholar in 2022, transitioning to a postdoctoral scholar after completing his doctoral studies. His PhD research centered on novel multiparametric systems, while concurrently investigating physiological mechanisms related to Ent-Vert applications. He currently contributes to clinical research on interventional transcatheter systems at Abbott, focusing on right-sided structural heart disease.

- You must start the Methods section with a paragraph headed [Ethical Approval](https://jp.msubmit.net/cgi-bin/main.plex?form_type=display_requirements#methods). If experiments were conducted on humans, confirmation that informed consent was obtained, preferably in writing, that the studies conformed to the standards set by the latest revision of the Declaration of Helsinki and that the procedures were approved by a properly constituted ethics committee, which should be named, must be included in the article file. If the research study was registered (clause 35 of the Declaration of Helsinki), the registration database should be indicated, otherwise the lack of registration should be noted as an exception (e.g. The study conformed to the standards set by the Declaration of Helsinki, except for registration in a database). For further information see: <https://physoc.onlinelibrary.wiley.com/hub/human-experiments>.

We thank the editor for bringing this to our attention. The Methods section has been updated to begin with a paragraph titled “Ethical Approval.” This section now explicitly states that the study was deemed Institutional Review Board (IRB) exempt by Northwestern University, in accordance with the ethical standards outlined in the Declaration of Helsinki.

- The reference list must be in alphabetical order, rather than numbered, to comply with our https://jp.msubmit.net/cgi-bin/main.plex?form_type=display_requirements#refs>Journal format.

We thank the editor for bringing this to our attention. The reference list has been reformatted in alphabetical order to conform to The Journal of Physiology reference style requirements.

- Please upload separate high-quality https://jp.msubmit.net/cgi-bin/main.plex?form_type=display_requirements#figures>figure files via the submission form.

- Please ensure that the Article File you upload is a Word file.

- Papers must comply with the Statistics Policy: https://jp.msubmit.net/cgi-bin/main.plex?form_type=display_requirements#statistics.

In summary:

- If n {less than or equal to} 30, all data points must be plotted in the figure in a way that reveals their range and distribution. A bar graph with data points overlaid, a box and whisker plot or a violin plot (preferably with data points included) are acceptable formats.

- If $n > 30$, then the entire raw dataset must be made available either as supporting information, or hosted on a not-for-profit repository, e.g. FigShare, with access details provided in the manuscript.

- 'n' clearly defined (e.g. x cells from y slices in z animals) in the Methods. Authors should be mindful of pseudoreplication.

- All relevant 'n' values must be clearly stated in the main text, figures and tables.

- The most appropriate summary statistic (e.g. mean or median and standard deviation) must be used. Standard Error of the Mean (SEM) alone is not permitted.

- Exact p values must be stated. Authors must not use 'greater than' or 'less than'. Exact p values must be stated to three significant figures even when 'no statistical significance' is claimed.

- Please include an Abstract Figure file, as well as the Figure Legend text within the main article file. The Abstract Figure is a piece of artwork designed to give readers an immediate understanding of the research and should summarise the main conclusions. If possible, the image should be easily 'readable' from left to right or top to bottom. It should show the physiological relevance of the manuscript so readers can assess the importance and content of its findings. Abstract Figures should not merely recapitulate other figures in the manuscript. Please try to keep the diagram as simple as possible and without superfluous information that may distract from the main conclusion(s). Abstract Figures must be provided by authors no later than the revised manuscript stage and should be uploaded as a separate file during online submission labelled as File Type 'Abstract Figure'. Please also ensure that you include the figure legend in the main article file. All Abstract Figures should be created using BioRender. Authors should use The Journal's premium BioRender account to export high-resolution images. Details on how to use and access the premium account are included as part of this email.

***ent-Vert* selectively inhibits RyR2-mediated sarcoplasmic reticulum Ca²⁺ leak and prevents triggered activity in human cardiac slices.** a) In human ventricular slices exposed to β -adrenergic stimulation with isoproterenol and RyR2 sensitization with caffeine, RyR2 channels become hyperactive, leading to Ca²⁺ sarcoplasmic leak and triggered electrical activity. b) *ent-Vert* binds to RyR2 and stabilizes its closed state, thereby reducing sarcoplasmic Ca²⁺ leak and suppressing triggered electrical activity. These findings demonstrate the translational potential of selective RyR2 inhibition as a targeted antiarrhythmic strategy for ventricular arrhythmias.

- Please include a full title page as part of your main article (Word) file, which should contain the following: title, authors, affiliations, corresponding author name and contact details, keywords, and running title.

We thank the editor for bringing this to our attention. The title page has been updated to include all required information, including the running title, full author list with affiliations, corresponding author details, and keywords.

Running Title: Selective RyR2 Inhibition Lowers Arrhythmia Susceptibility

- Please ensure that all figures and tables have a title and legend, and that they have been cited within the main article text.

We thank the editor for noting this. All figures and tables now include complete titles and legends, and each has been explicitly cited in the main text. In particular, Figures 3E and 3F have been added to the Results section for clarity.

EDITOR COMMENTS

Reviewing Editor: Ethics Concerns:

This study involves human donor hearts that were not used for transplantation, and the study protocol, submitted to local (Northwestern University's) Institutional Review Board (IRB), was deemed IRB exempt. It might not be applicable/possible to have written consents in this case.

An appropriate reference number for the approval is not stated.

Thank you for bringing this to our attention. While this study was deemed IRB exempt by Northwestern University, we have provided the IRB Study Reference Number STU00216867.

Comments to the Author:

The manuscript has been reviewed by experts in this field. The major strengths of the manuscript include testing whether selective RyR2 inhibition is effective against arrhythmias, and use of human cardiac tissue. However, the reviewers also noted several weaknesses that require attention. These pertain to the arrhythmia induction protocol relying on caffeine, lack of experimental data at the cardiomyocyte level, some aspects of data interpretation, and statistical approaches and power. The reviewing editor concurs with these assessments.

REFEREE COMMENTS

Referee #1:

Madrid et al have investigated the effects of ent-verticilide on some factors underpinning arrhythmias. They conclude that the drug suppresses arrhythmic triggers and reduces arrhythmogenic substrates. Unlike dantrolene, verticilide does not affect skeletal muscle potentially making it an attractive choice as an antiarrhythmic.

A major strength of the work is that it is carried out on human cardiac tissue. There are, however, several weaknesses as described below.

1. Ectopy is induced by adding iso together with caffeine. The use of caffeine means that any arrhythmias arise, at least in part, from increasing the open probability of the RyR, This makes it less surprising that a drug which decreases RyR2 opening can decrease ectopy. It would have been more interesting if the arrhythmias had been provoked without interfering with the RyR.

We acknowledge that our current study evaluates ent-Verticilide in an isoproterenol + caffeine-induced arrhythmia model rather than intrinsically diseased human tissue. We selected this model because it specifically provokes RyR2-dependent Ca^{2+} release abnormalities, a major driver of triggered activity in multiple cardiac pathologies. Caffeine increases RyR2 open probability, while isoproterenol enhances SR Ca^{2+} loading through β -adrenergic stimulation, together unmasking RyR2-mediated afterdepolarizations. This combination thus provides a sensitive and mechanistically appropriate assay for testing a compound that selectively stabilizes the closed state of RyR2 and reduces Ca^{2+} leak. Moreover, we employed an established preclinical protocol for assessing RyR2-mediated Ca^{2+} leak using isoproterenol and caffeine in our previous publications: George et al., Am. J. Physiol., 2023, Batiste et al., PNAS, 2019.

2. While it is good that you test the data for normality before using parametric tests, it should be noted that such tests have little power to reveal deviations from normality with the relatively small sample sizes used.

We appreciate the reviewer's observation regarding the limitations of normality testing with small sample sizes. We acknowledge that the Shapiro-Wilk test has reduced power to detect deviations from normality with $n = 4 - 5$. Given this limitation and the small sample size, we selected statistical approaches that are robust to violations of normality assumptions. For the ectopy incidence data (Figure 2c), we used the Friedman test (a non-parametric test for repeated measures) and have now applied the Wilcoxon signed-rank tests for pairwise post-hoc comparisons, as suggested. The Wilcoxon test does not assume normality or identical distributions and is suitable even when the baseline condition has zero variance.

For APD and CV_T data (Figures 3 and 4), which showed less deviation from normality, we used repeated measures ANOVA/Mixed Effects models. These methods are generally robust to moderate departures from normality. Additionally, the use of within-subject design reduces variability and increases the reliability of these tests even in small samples. Our conservative approach of defaulting to non-parametric methods when normality was uncertain (as with ectopy data) ensures that our statistical conclusions are valid and not dependent on distributional assumptions.

3. You use Dunn's post hoc test in Fig 2C. As discussed in the original paper (Dunn, 1964) this assumes that the populations are identically distributed. This is not the case in your data where there is zero spread in the baseline condition. I suggest that you check with a statistician whether this may invalidate the apparent significance of your result.

We sincerely thank the reviewer for this comment and fully acknowledge that Dunn's post-hoc test assumes identical distribution. As noted, this assumption is violated in our dataset due to zero variance in the baseline condition, which renders Dunn's test inappropriate.

To address this, we have now replaced Dunn's test with Wilcoxon signed-rank tests as post-hoc comparisons. The Wilcoxon signed-rank test is a non-parametric method suited for comparing paired or repeated measures data without assuming normality or equal variances (Wilcoxon, 1945). It focuses on ranking the absolute differences between paired observations (e.g., baseline vs. treatment condition), assigning signs based on the direction of change, and testing whether the median difference is zero. In our case, where baseline values are uniformly zero, the differences simplify to the post-baseline values themselves, allowing the test to rank and evaluate these non-zero changes. To evaluate if ent-Vert 3 uM significantly reduces ectopy incidence using the Wilcoxon signed-rank test, we achieve a trending p-value of 0.0625 showing that *ent-Vert* has a trend toward lowering Iso+caffeine-induced ectopy.

4. In Fig 2C there do not appear to be 4 points in each of the conditions. I assume that some are superimposed but this needs to be made clearer, perhaps by separating horizontally.

Thank you for bringing this to our attention. You are correct that some data points in Figure 2C were previously colocalized. We have revised the figure to slightly offset the points horizontally, allowing all four data points in each condition to be clearly visualized.

5. Lines 220-222. You state that in some hearts, ectopy was observed in the LV but not in the RV. However, the two upper panels of Fig 2C show 4 points with ectopy in the LV and 3 in the RV. This doesn't seem like a solid base from which to infer chamber differences. Is there any statistical basis to the statement?

We thank the reviewer for this helpful comment. The reviewer is correct that this observation was not supported by statistical analysis. The statement was based solely on qualitative observations. To avoid overinterpretation, we have removed the sentence "highlighting ventricle-specific differences despite the paired sampling" from the manuscript.

6. Lines 277-280. Here you compare the lack of effect of ent-Vert which does not affect APD

with dantrolene. You quote your previous work (Geroge et al) as showing that dantrolene "tends to prolong APD in LV slices". It does not seem rigorous to compare non-significant effects of two drugs in this way.

We thank the reviewer for this helpful comment. We agree that a direct statistical comparison between two non-significant effects is not rigorous. Our intent was only to provide qualitative context regarding previously observed electrophysiological trends with dantrolene. To clarify this, we have revised the sentence to avoid implying a direct comparison and to emphasize that the reference to dantrolene serves as contextual background rather than a statistical contrast.

7. Given the lack of effect on APD, why do you conclude (line 53) that ent-Vert "reduced arrhythmogenic substrates".

We thank the reviewer for this thoughtful comment. In this context, we used the term arrhythmogenic substrate to refer to the propensity for triggered activity arising from aberrant Ca^{2+} handling rather than from altered repolarization. Although *ent-Vert* did not affect APD, it significantly reduced the incidence of ectopic events induced by Iso+caffeine, indicating stabilization of RyR2-mediated Ca^{2+} release and suppression of afterdepolarizations. We have revised the text to clarify that the reduction in arrhythmogenic substrate refers specifically to the mitigation of Ca^{2+} -driven triggers rather than changes in repolarization dynamics.

Referee #2:

The paper addresses one important aspect, namely whether selective RyR2 inhibition is effective against arrhythmias. The used novel and highly selective RyR2 inhibitor was shown to be effective in genetically modified mice with an increased susceptibility to arrhythmias. Since this cannot be directly translated to the human, such studies and the present one are essential for successful clinical translation.

The authors show in human ventricular slices that ventricular ectopy induced by ISO+caffeine is significantly reduced by the RyR2 inhibitors. CV and APD remained unchanged. The latter was interpreted as justification for the claim that only focal ectopic activity, but reentry termination is the potential reason for the antiarrhythmic effect of the RyR2 inhibitor.

I have the following comments:

The author studied the effects in tissue. How sure are you that the penetration of the drug was sufficient enough to allow to conclude that CV and APD are reliably UNchanged? Could APD recordings be repeated in isolated human ventricular myocytes?

We appreciate the reviewer's thoughtful comment regarding drug penetration in our human cardiac slice preparations. We are confident that *ent-Vert* diffusion was sufficient across the 400- μ m-thick tissue based on several lines of evidence.

First, the dose-dependent suppression of ectopic activity (Figure 2C) indicates that *ent-Vert* effectively reaches and inhibits RyR2 channels throughout the slice, as ectopic foci can arise from any intramural locations. Second, our laboratory and others have previously demonstrated adequate penetration of pharmacological agents, including the RyR2 inhibitor dantrolene, in human ventricular slices of comparable thickness (George et al., Am. J. Physiol., 2023). Together, these findings support that the absence of changes in conduction velocity (CV) and action-potential duration (APD) reflects a true lack of electrophysiological effect rather than limited drug diffusion.

We agree that studies in isolated human ventricular cardiomyocyte could serve as a valuable complementary approach. However, obtaining viable isolated human ventricular myocytes is technically challenging, and such cells often exhibit altered morphology, electrophysiology, and calcium handling, and lack cell-cell coupling that do not reliably reflect native cardiac function. In contrast, human organotypic cardiac slice preparations preserve the native multicellular architecture, cell-cell coupling, and extracellular matrix, thereby providing a more physiologically relevant model for assessing drug effects on human cardiac electrophysiology.

The effect of the RyR2 inhibitor on RyR2 function is rather indirect. Any chance to show more direct effects (RyR2 single channel recordings or CaT properties, SR Ca²⁺ leak at the single cardiomyocyte level).

We appreciate the reviewer's interest in more direct measurements of RyR2 function. The direct effects of *ent-Vert* on RyR2 function have been comprehensively characterized in our previous publication in PNAS (Batiste et al., PNAS, 2019), where detailed analyses of isolated cardiomyocytes demonstrated that *ent-Vert* selectively inhibits RyR2-mediated Ca²⁺ leak. These studies included single-cell assessments of Ca²⁺ sparks, spontaneous Ca²⁺ release, SR Ca²⁺ leak, and Ca²⁺ transient properties, confirming RyR2 selectivity without effects on other Ca²⁺-handling proteins such as RyR1, SERCA2a, or L-type Ca²⁺ channels. Furthermore, our subsequent study, now in press (Do et al., Molecular Pharmacology, 2025), directly examined RyR2 single-channel recordings in artificial lipid bilayers and demonstrated that *ent-Vert* directly inhibits RyR2 by increasing mean closed time without altering mean open time, thereby stabilizing the closed-channel state. This work provides the first detailed mechanistic evidence of *ent-Vert*'s direct modulation of RyR2 gating and defines its structure-activity relationship through comparison with a novel analog (activert) that conversely functions as a RyR2 activator.

The current study builds upon this established mechanistic foundation to evaluate the translational potential of *ent-Vert* in human cardiac tissue under conditions that simulate arrhythmogenic triggers associated with structural heart disease. Our focus was to demonstrate that RyR2 inhibition suppresses arrhythmogenic triggers in human tissue without altering fundamental electrophysiological parameters (APD, CV_T) that are critical for therapeutic applications. We believe the combination of rigorous mechanistic studies in prior publications and the present translational findings in human tissue provides compelling evidence for *ent-Vert*'s selective RyR2 inhibition and antiarrhythmic potential.

The effects shown with this new drug are highly variable in Fig. 2, with absence of effect in some preparations. Could this be a bias? Did the authors calculate the required sample size to reach reliable statistical conclusions?

Variability: We thank the reviewer for this important comment. The observed variability in Figure 2 reflects the inherent biological heterogeneity of human cardiac tissue, which represents diverse genetic backgrounds and clinical histories, unlike inbred animal models or homogeneous cell cultures. We have revised Figure 2c to better visualize the individual data points for each heart, providing a transparent presentation of inter-preparation variability.

Sample size calculation: We performed a power analysis assuming a large effect size (Cohen's $f = 0.8$), which is appropriate given that *ent-Vert* is intended to substantially reduce ectopic activity incidence (a robust measure of arrhythmogenic triggers). With $n = 4$ hearts in a repeated measures design (4 conditions per heart), we achieved 87% power to detect such effects at $\alpha = 0.05$. This effect size assumption is consistent with our prior work (George et al., 2023) and reflects the magnitude of effect expected when a pharmacological intervention successfully suppresses arrhythmogenic activity. The paired, within-subjects design maximizes statistical efficiency while accommodating the scarcity of viable human cardiac tissue, which provides direct translational relevance that cannot be achieved with animal models.

Dear Professor Efimov,

Re: JP-RP-2025-290283R1 "**Selective RyR2 Inhibition Reduces Arrhythmia Susceptibility in Human Cardiac Slices**"
by Micah Madrid, Katy Trampel, Batool Salman, Sharon A George, Tatiana Efimova, Bjorn C. Knollmann, and Igor R. Efimov

We are pleased to tell you that your paper has been accepted for publication in The Journal of Physiology.

Yours sincerely,

Natalia Trayanova
Senior Editor
The Journal of Physiology

IMPORTANT POINTS TO NOTE FOLLOWING ACCEPTANCE OF YOUR PAPER:

- **IMPORTANT NOTICE ABOUT OPEN ACCESS:** To assist authors whose funding agencies mandate immediate public access to published research findings, The Journal of Physiology allows authors to pay an Open Access (OA) fee to have their papers made freely available immediately on publication.

- You can help your research get the attention it deserves! Check out Wiley's free Promotion Guide for best-practice recommendations for promoting your work at: www.wileyauthors.com/eeo/guide. You can learn more about Wiley Editing Services which offers professional video, design, and writing services to create shareable video abstracts, infographics, conference posters, lay summaries, and research news stories for your research at: www.wileyauthors.com/eeo/promotion.

- If you would like to receive our 'Research Roundup', a monthly newsletter highlighting the cutting-edge research published in The Physiological Society's family of journals (The Journal of Physiology, Experimental Physiology, Physiological Reports, The Journal of Nutritional Physiology and The Journal of Precision Medicine: Health and Disease), please click this link, fill in your name and email address and select 'Research Roundup': <https://www.physoc.org/journals-and-media/membernews>

EDITOR COMMENTS

Reviewing Editor:

Many thanks for the revision. The reviewers and reviewing editor do not have further comments.

REFEREE COMMENTS

Referee #1:

The authors have responded constructively to my comments

Referee #2:

Thank you for responsive revision. No further comments.